:ꞎ: PLOS | ONE

# Awareness of polycystic ovary syndrome among obstetrician-gynecologists and endocrinologists in Northern Europe

Terhi T. Piltonen[1]\*, Maria Ruokojärvi[1], Helle Karro[2], Linda Kujanpää[1], Laure Morin-Papunen[1], Juha S. Tapanainen[1,3], Elisabet Stener-Victorin[4], Inger Sundrström-Poromaa[5], Angelica L. Hirschberg[6], Pernille Ravn[7], Dorte Glintborg[8], Jan Roar Mellembakken[9], Thora Steingrimsdottir[10], Melanie Gibson-Helm[11], Eszter Vanky[12,13], Marianne Andersen[14], Riikka K. Arffman[1], Helena Teede[11], Kobra Falah-Hassani[1]

1 Department of Obstetrics and Gynaecology, University of Oulu and Oulu University Hospital, Medical Research Centre, PEDEGO Research Unit, Oulu, Finland, 2 Department of Obstetrics and Gynaecology, Faculty of Medicine, University of Tartu, Tartu, Estonia, 3 Department of Obstetrics and Gynaecology, University of Helsinki and Helsinki University Hospital, Helsinki, Finland, 4 Department of Physiology and Pharmacology, Karolinska Institutet, Stockholm, Sweden, 5 Department of Women's and Children's Health, Uppsala University, Uppsala, Sweden, 6 Department of Women's and Children's Health, Karolinska Institutet and Department of Gynecology and Reproductive Medicine, Karolinska University Hospital, Stockholm, Sweden, 7 Department of Gynaecology and Obstetrics, Odense University Hospital, Odense, Denmark, 8 Department of Endocrinology, Odense University Hospital, Odense, Denmark, 9 Department of Reproductive Medicine, Division of Gynaecology and Obstetrics, Oslo University Hospital, Oslo, Norway, 10 Department of Obstetrics and Gynaecology, Landspitali University Hospital, School of Health Sciences, Faculty of Medicine, University of Iceland, Reykjavik, Iceland, 11 Monash Centre for Health Research and Implementation, School of Public Health and Preventive Medicine, Monash University, Melbourne, Australia, 12 Department of Clinical and Molecular Medicine, Norwegian University of Science and Technology, Trondheim, Norway, 13 Department of Obstetrics and Gynaecology, St Olav's Hospital, University Hospital of Trondheim, Trondheim, Norway, 14 Department of Language and Culture, UiT—The Arctic University of Norway, Tromsø, Norway

\* terhi.piltonen@oulu.fi

**Data Availability Statement:** All relevant data are within the manuscript.

## Abstract

### Objective

To date, little is known about differences in the knowledge, diagnosis making and treatment strategies of health care providers regarding polycystic ovary syndrome (PCOS) across different disciplines in countries with similar health care systems. To inform guideline translation, we aimed to study physician reported awareness, diagnosis and management of PCOS and to explore differences between medical disciplines in the Nordic countries and Estonia.

### Methods

This cross-sectional survey was conducted among 382 endocrinologists and obstetrician-gynaecologists in the Nordic countries and Estonia in 2015–2016. Of the participating physicians, 43% resided in Finland, 18% in Denmark, 16% in Norway, 13% in Estonia, and 10% in Sweden or Iceland, and 75% were obstetrician-gynaecologists. Multivariable logistic

**Funding:** This study was supported by the Sigrid Juselius foundation, the Academy of Finland (grant no: 321763) and the Finnish Medical Association.

**Competing interests:** The authors have declared that no competing interests exist.

regression models were run to identify health care provider characteristics for awareness, diagnosis and treatment of PCOS.

## Results

Clinical features, lifestyle management and comorbidity were commonly recognized in women with PCOS, while impairment in psychosocial wellbeing was not well acknowledged. Over two-thirds of the physicians used the Rotterdam diagnostic criteria for PCOS. Medical endocrinologists more often recommended lifestyle management (OR = 3.6, CI 1.6–8.1) or metformin (OR = 5.0, CI 2.5–10.2), but less frequently OCP (OR = 0.5, CI 0.2–0.9) for non-fertility concerns than general obstetrician-gynaecologists. The physicians aged <35 years were 2.2 times (95% CI 1.1–4.3) more likely than older physicians to recommend lifestyle management for patients with PCOS for fertility concerns. Physicians aged 46–55 years were less likely to recommend oral contraceptive pills (OCP) for patients with PCOS than physicians aged >56 (adjusted odds ratio (OR) = 0.4, 95% CI 0.2–0.8).

## Conclusion

Despite well-organized healthcare, awareness, diagnosis and management of PCOS is suboptimal, especially in relation to psychosocial comorbidities, among physicians in the Nordic countries and Estonia. Physicians need more education on PCOS and evidence-based information on Rotterdam diagnostic criteria, psychosocial features and treatment of PCOS, with the recently published international PCOS guideline well needed and welcomed.

## Introduction

Polycystic ovary syndrome (PCOS) is the most common endocrine disorder among women of fertile age [1]. The prevalence of PCOS ranges between 5% and 16%, depending on the ethnic groups and diagnostic criteria [2, 3]. Recent diagnostic criteria include the original National Institutes of Health (NIH), the Androgen Excess Society (AE-PCOS Society) and the new internationally endorsed Rotterdam criteria [2, 3], all of which take into account only repro-ductive health features such as oligo-ovulation or anovulation, hyperandrogenism, and poly-cystic ovaries. However, PCOS affects not only the women's sexual and reproductive health, but also their metabolic health and psychological wellbeing [4–7]. To date, the symptoms and features included in the Rotterdam criteria (oligomenorrhea, hirsutism/biochemical hyperan-drogenism, polycystic ovaries) as well as metabolic features associated with PCOS are recog-nised by medical doctors internationally, whereas doctors are less aware of psychological comorbidity, such as anxiety and depression [8–10]. This leaves room for improvement of the awareness of comorbidities linked to PCOS, especially the psychological ones.

Previously, we reported differences in the diagnosis and treatment of PCOS across coun-tries and between endocrinologists and obstetrician-gynecologists [10]. In Europe, around three-quarters of obstetrician-gynecologists and endocrinologists use the Rotterdam criteria, while in North America approximately half of these health professionals use the Rotterdam cri-teria, preferring the NIH criteria [10]. Moreover, endocrinologists are more likely to use the Rotterdam diagnostic criteria than obstetrician-gynecologists [10]. Reproductive and medical

endocrinologists, on the other hand, are more likely to recommend lifestyle changes for the management of PCOS than obstetrician-gynecologists [10].

The aggregated results from many European countries on awareness and management of PCOS [10] cannot be generalized to the Nordic countries. There are a wide range of different health care systems in Europe. However, the Nordic countries (Finland, Denmark, Norway, Sweden and Iceland) and Estonia share similar health care systems [11]. Access to healthcare is high in these counties [12] and they are among countries with more equal distribution of income and have similarity in some lifestyle risk factors such as obesity [13]. To date, differences in the knowledge, diagnosis and treatment of PCOS across the Nordic countries among obstetrician-gynecologists and endocrinologists are not known. In the context of the new international guidelines for the diagnosis and management of PCOS, it is important to establish baseline practice and identify areas for improvement and translation. We aimed to study the awareness, diagnosis and management of PCOS and to determine the differences in physician characteristics in the Nordic countries and Estonia.

## Materials and methods

### Study population

This cross-sectional survey was conducted among medical and reproductive endocrinologists and obstetrician-gynecologists in 2015–2016. The survey questionnaire is available online [8] and was part of larger international study [10] conducted to inform translation needs for the new international PCOS guidelines that were published in 2018 [14]. The questionnaire and methods of the larger study have been described in detail previously. The survey questionnaire was adapted from the questionnaires used to collect data from physicians in Australia [9] and Europe [15]. The present data was partly included in the broader European group of the international study [10], but was not disaggregated by region (e.g., Scandinavia). We also added new data from Iceland for the analysis. In the current analysis, we report the results for each of the five Nordic countries and Estonia as well as the results for all the Nordic countries and Estonia combined. Given the European Union regulations on individual data sharing, the distribution of the link to access the questionnaire was done through the national societies (except for Iceland), i.e. the Finnish Society of Obstetrics and Gynecology, Finnish Society of Endocrinology, Danish Society of Endocrinology, Danish Society of Obstetrics and Gynecology, Norwegian Society for Gynecology and Obstetrics, Norwegian Society of Endocrinology, Estonian Gynecologists' Society. These medical societies sent an e-mail invitation to the physicians and provided the link to the questionnaire. However, the Swedish Society of Obstetrics and Gynecology did not send a personal e-mail invitation to physicians but announced the study and provided the link to the questionnaire on their website. Icelandic members of the Nordic PCOS Network identified the specialists and e-mail invitations to access the link to the questionnaire were sent through them. In the beginning of the questionnaire was a short introduction announcing that the questionnaire was sent on behalf of the Nordic PCOS network and that the questionnaire should only be replied once. The Ethical Committee of Oulu University Hospital, Oulu, Finland approved the study. Participation in this study was voluntary and the responses were given anonymously. If the participant did not report being an obstetrician-gynecologist or endocrinologist, the answers were excluded.

### Independent and dependent variables

Information on nationality, age, gender, specialty, PCOS diagnostic criteria (the Rotterdam, NIH, AE-PCOS Society, or other) [2, 3], approximate number of women with PCOS cared for in last year, approximate national prevalence of PCOS, PCOS clinical features, psychological

and psychosocial factors related to PCOS, comorbidities related to PCOS, mode of support for PCOS, and lifestyle management for PCOS was gathered by a questionnaire. The questionnaire was carried out in English.

## Statistical analysis

We first tested differences in physician characteristics, clinical features of PCOS, common reasons for clinic attendance, important long-term concerns, psychosocial wellbeing and comorbidities associated with PCOS, lifestyle management of PCOS, mode of support and treatment of PCOS between the countries using chi-square test. We then ran multivariable logistic regression models to identify health care provider characteristics for the following nine most important outcomes: 1) awareness of symptom improvement with weight loss, 2) estimated national PCOS prevalence, 3) using Rotterdam diagnostic criteria, 4) recommending oral contraceptive pills (OCP), 5) recommending clomiphene citrate, 6) recommending metformin plus clomiphene citrate, 7) recommending lifestyle management for non-fertility concerns, 8) recommending metformin for non-fertility concerns, and 9) recommending lifestyle management for fertility concerns. We used Stata, version 15 (StataCorp, College Station, TX) for the analyses.

## Results

### Participant characteristics

The characteristics of the participants per country are presented in Table 1. A total 382 participants were included in the analyses. Of participating physicians, 43.2% resided in Finland, 17.8% in Denmark, 16.0% in Norway, 12.8% in Estonia, 6.5% in Sweden and 3.7% in Iceland. Seventy-five percent of the participants were obstetrician-gynecologists and 25% were endocrinologists, and 79% were women. Twenty-eight percent of the physicians reported seeing more than 50 women with PCOS per year and 43% estimated the national prevalence of PCOS to be more than 10%. Over two-thirds of the physicians used the Rotterdam criteria for diagnosing PCOS.

### Clinical features, psychosocial wellbeing, lifestyle management and comorbidities

Irregular menstrual cycle was most commonly reported clinical feature (Table 2 and Fig 1). In line with this, infertility was the most frequent reason for clinic attendance for PCOS in all the Nordic countries and Estonia. The second most common reason for clinic attendance was obesity and type 2 diabetes (Table 3). Scalp hair loss was the least reported feature of PCOS (Table 2).

Tendency to gain weight and trouble losing weight in affected women was commonly recognized as well as the effect of weight loss and exercise on PCOS symptoms. The most commonly reported comorbidities were reduced fertility, type 2 diabetes, gestational diabetes, insulin resistance, and cardiovascular disease risk factors. Compared to other features related to PCOS, the reduction of psychosocial wellbeing in PCOS was less recognized by the health professionals. Indeed, depression and especially anxiety were commonly ranked low in the context of psychosocial features of PCOS. On the other hand, reduced quality of life was most commonly reported in Denmark, Finland and Estonia, while body image dissatisfaction was most commonly reported in Iceland, Norway and Sweden. Fatty liver, sleep apnea, pregnancy complications and risk for endometrial cancer were less commonly known features. Physicians in Finland were more aware of risk for fatty liver in women with PCOS compared with

**Table 1. The characteristics of the study population by country, proportions (%).**

| Characteristic | Overall (N = 382) | Denmark (N = 68) | Estonia (N = 49) | Finland (N = 165) | Iceland (N = 14) | Norway (N = 61) | Sweden (N = 25) | P |
|---|---|---|---|---|---|---|---|---|
| *Sex* | | | | | | | | |
| Men | 21 | 30 | 8 | 15 | 43 | 30 | 32 | 0.001 |
| Women | 79 | 70 | 92 | 85 | 57 | 70 | 68 | |
| *Age* | | | | | | | | |
| ≥35 | 17 | 19 | 33 | 15 | 0 | 18 | 12 | 0.005 |
| 36–45 | 30 | 40 | 16 | 30 | 29 | 30 | 24 | |
| 46–55 | 25 | 15 | 29 | 25 | 21 | 36 | 16 | |
| ≥56 | 28 | 26 | 22 | 30 | 50 | 16 | 48 | |
| *Specialty* | | | | | | | | |
| OBGYN | 75 | 55 | 100 | 76 | 93 | 79 | 60 | <0.001 |
| RE | 10 | 4 | 0 | 15 | 7 | 2 | 32 | |
| ME | 15 | 41 | 0 | 9 | 0 | 19 | 8 | |
| *No. of women with PCOS cared for in last year* | | | | | | | | |
| <50 | 72 | 78 | 88 | 70 | 71 | 68 | 56 | <0.001 |
| 50–200 | 26 | 22 | 10 | 30 | 29 | 25 | 32 | |
| >200 | 2 | 0 | 2 | 0 | 0 | 7 | 12 | |
| *Approximate prevalence of PCOS* | | | | | | | | |
| 0–10% | 57 | 48 | 71 | 58 | 57 | 59 | 48 | 0.21 |
| 11–20% | 43 | 52 | 29 | 42 | 43 | 41 | 52 | |
| *Diagnosis criteria used (N = 374)* | | | | | | | | |
| National Institutes of Health | 3 | 0 | 12 | 4 | 0 | 0 | 0 | <0.001 |
| Rotterdam | 69 | 79 | 43 | 60 | 93 | 85 | 92 | |
| AE and PCOS Society | 2 | 3 | 8 | 0 | 0 | 2 | 0 | |
| Do not know | 23 | 13 | 33 | 33 | 7 | 13 | 4 | |
| Other * | 3 | 5 | 4 | 3 | 0 | 0 | 4 | |

ME, medical endocrinologist; OBGYN, obstetrician-gynaecologist; PCOS, polycystic ovary syndrome; RE, reproductive endocrinologist

* Included the official diagnostic criteria or national guidelines of different countries

physicians in other Nordic countries and Estonia. Physicians in Norway and Iceland reported pregnancy complications more commonly than in other countries. The doctors were generally well informed that ovarian cancer is not related to PCOS (Fig 1). Sixteen percent of the participants reported an association between surgery for ovarian cysts and PCOS (Fig 1). There were no differences between the countries. Fifty-eight percent of the participants thought PCOS is underdiagnosed and 23% thought it is overdiagnosed.

The physicians ranked long-term health concerns related to PCOS as obesity, type 2 diabetes, infertility and cardiovascular diseases most important, whereas psychological wellbeing and endometrial cancer were not ranked important (Table 3).

## Treatment of PCOS

OCP and lifestyle modifications were the most commonly prescribed treatments for non-fertility concern in all the countries, except Estonia, where metformin was the second most commonly prescribed treatment after OCP for non-fertility concern (Table 4). For fertility concern, lifestyle modification was the most commonly prescribed treatment in Denmark, Estonia, Finland and Norway, and ovulation inductors in Iceland and Sweden.

**Table 2. The differences in Nordic countries' and Estonia's health professionals' views on clinical features, psychosocial wellbeing, lifestyle management and comorbidities associated with PCOS.** The estimates are proportions (%).

| Characteristic | Overall (N = 382) | Denmark (N = 68) | Estonia (N = 49) | Finland (N = 165) | Iceland (N = 14) | Norway (N = 61) | Sweden (N = 25) | P |
|---|---|---|---|---|---|---|---|---|
| *Clinical features (N = 382)* | | | | | | | | |
| Irregular menstrual cycles | 98 | 99 | 93.9 | 99 | 100 | 100 | 96 | 0.08 |
| Excess hair growth | 88 | 97 | 75.5 | 84 | 93 | 97 | 92 | 0.001 |
| Scalp hair loss | 51 | 59 | 44.9 | 42 | 64 | 71 | 48 | 0.003 |
| High blood androgen levels | 93 | 99 | 85.7 | 93 | 93 | 92 | 88 | 0.16 |
| Acne | 87 | 94 | 75.5 | 86 | 79 | 95 | 92 | 0.013 |
| *Psychosocial wellbeing (N = 382)* | | | | | | | | |
| Reduced quality of life | 63 | 78 | 51 | 59 | 79 | 66 | 56 | 0.02 |
| Depression | 42 | 57 | 33 | 36 | 43 | 46 | 40 | 0.05 |
| Anxiety | 24 | 32 | 18 | 21 | 57 | 20 | 28 | 0.01 |
| Body image dissatisfaction | 58 | 63 | 50 | 50 | 86 | 74 | 64 | 0.001 |
| *Lifestyle management (N = 382)* | | | | | | | | |
| Increased tendency for weight gain | 84 | 77 | 86 | 82 | 100 | 89 | 88 | 0.20 |
| Difficulty losing weight | 79 | 85 | 71 | 75 | 79 | 82 | 92 | 0.15 |
| Improvement of symptoms after weight loss | 84 | 93 | 61 | 84 | 100 | 90 | 88 | <0.001 |
| Improvement of symptoms with exercise | 60 | 81 | 33 | 56 | 64 | 67 | 68 | <0.001 |
| Improvement of symptoms with a low glycemic index diet | 42 | 46 | 39 | 39 | 64 | 51 | 20 | 0.05 |
| *Comorbidities (N = 382)* | | | | | | | | |
| Reduced fertility | 96 | 94 | 92 | 98 | 100 | 97 | 88 | 0.17 |
| Insulin resistance | 97 | 97 | 94 | 99 | 100 | 97 | 96 | 0.50 |
| Increased risk of type 2 diabetes | 95 | 96 | 86 | 96 | 100 | 95 | 92 | 0.08 |
| Increased risk of gestational diabetes | 83 | 74 | 78 | 89 | 79 | 87 | 72 | 0.03 |
| Increased risk of cardiovascular disease risk factors | 83 | 77 | 76 | 89 | 79 | 84 | 80 | 0.16 |
| Endometrial cancer | 54 | 47 | 45 | 55 | 79 | 53 | 72 | 0.12 |
| Fatty liver | 36 | 40 | 16 | 50 | 21 | 21 | 24 | <0.001 |
| Pregnancy complications | 53 | 49 | 47 | 49 | 79 | 74 | 44 | 0.004 |
| Sleep apnea and snoring | 34 | 31 | 27 | 38 | 29 | 41 | 20 | 0.26 |

## Multivariable regression analysis

Female physicians were 2.6 times more likely to estimate the national prevalence of PCOS more than 10% than male physicians (Table 5). Physicians aged ≤35 years were twice more likely to estimate the national prevalence of PCOS more than 10% than physicians aged ≥56 (Table 5). The physicians aged ≤35 years also 2.2 times more often recommended lifestyle management for patients with PCOS for fertility concerns than older physicians.

Physicians aged 46–55 years were less likely to recommend OCP for patients with PCOS than physicians aged ≥56. Medical endocrinologists more commonly recommended lifestyle management or metformin for PCOS for non-fertility concerns than obstetrician-gynecologists or reproductive endocrinologists. Physicians who treated more than 50 patients with PCOS annually, reported the national prevalence of PCOS >10% 2.5 times more frequently and used the Rotterdam diagnostic criteria three times more frequently than physicians who treated less than 50 patients with PCOS annually.

## Mode of support

Table 3 shows the health professionals views on mode of support that should be offered. The most common and least common modes of support were considered similar across all

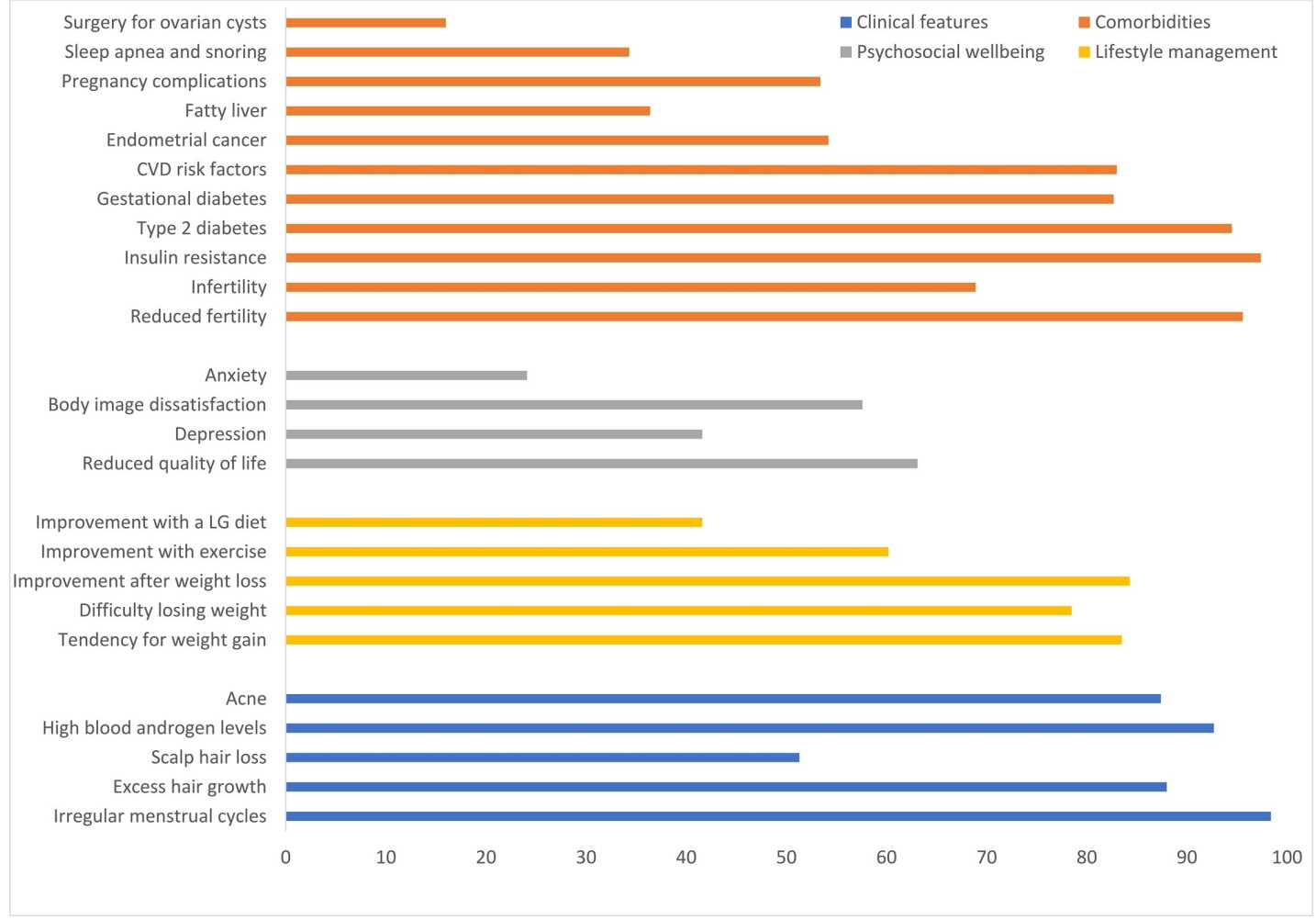

**Fig 1. The Nordic countries health professionals' views on clinical features, comorbidities, psychosocial wellbeing and lifestyle management associated with PCOS.** The estimates are proportions (%).

countries; the most needed mode of support was broadly available educational materials for health professionals and the least common mode was a regular email update. A need for PCOS specific website for health professionals was not ranked high especially in Finland compared to other countries.

## Discussion

### Main findings

This is the first study assessing PCOS awareness in health professionals across Nordic countries and Estonia. Over two-thirds of the physicians who answered the questionnaire in the Nordic countries and Estonia use the Rotterdam diagnostic criteria for PCOS. Clinical features, lifestyle management and comorbidities are commonly recognized in women with PCOS, while the reduction of psychosocial wellbeing is less associated with PCOS. Infertility is the most frequent reason and obesity and type 2 diabetes the second most common reason for clinic attendance for PCOS in the Nordic countries and Estonia. There are some differences in the treatments for PCOS between physicians in the Nordic countries and Estonia even though

**Table 3. The differences in Nordic countries' and Estonia's health professionals' views on most common reason for clinic attendance, most important long-term concern about PCOS, and mode of support.** The estimates are proportions (%).

| Health professionals' views | Overall | Denmark | Estonia | Finland | Iceland | Norway | Sweden | P |
|---|---|---|---|---|---|---|---|---|
| The most common reason for clinic attendance (N = 378) | | | | | | | | |
| Infertility | 77.3 | 70.6 | 95.9 | 77.3 | 69.2 | 75.4 | 66.7 | 0.019 |
| Cardiovascular diseases | 0.8 | 1.5 | 2.0 | 0.6 | 0 | 0 | 0 | 0.81 |
| Obesity and type 2 diabetes | 13.8 | 20.6 | 16.3 | 14.1 | 0 | 9.8 | 4.2 | 0.17 |
| Endometrial cancer | 1.3 | 0 | 2.0 | 2.5 | 0 | 0 | 0 | 0.54 |
| Psychosocial problems | 2.7 | 2.9 | 2.0 | 0.6 | 0 | 8.2 | 4.2 | 0.061 |
| The most important long-term concern about PCOS (N = 380) | | | | | | | | |
| Infertility | 15.5 | 10.5 | 36.7 | 15.2 | 7.7 | 11.5 | 4.0 | 0.001 |
| Cardiovascular diseases | 12.9 | 9.0 | 6.1 | 17.0 | 15.4 | 11.5 | 12.0 | 0.35 |
| Obesity and type 2 diabetes | 63.4 | 71.6 | 46.9 | 62.4 | 76.9 | 63.9 | 72.0 | 0.088 |
| Endometrial cancer | 5.8 | 6.0 | 8.2 | 4.9 | 0 | 6.6 | 8.0 | 0.86 |
| Psychosocial problems | 1.3 | 1.5 | 0 | 0 | 0 | 4.9 | 4.0 | 0.062 |
| Mode of support (N = 379) | | | | | | | | |
| Broadly available educational materials for HPs | 81 | 77 | 76 | 81 | 93 | 83 | 80 | 0.71 |
| Presentation at HP forums and workshops | 58 | 44 | 57 | 59 | 64 | 67 | 68 | 0.12 |
| A PCOS website for HPs | 50 | 52 | 57 | 37 | 64 | 65 | 68 | 0.001 |
| A regular email update for HPs | 28 | 32 | 33 | 22 | 29 | 37 | 24 | 0.25 |
| Resources for women with PCOS | 57 | 53 | 65 | 51 | 86 | 65 | 56 | 0.06 |

HPs, health professionals

the countries share similar health care systems. Younger physicians more often recommend lifestyle management for patients with PCOS for fertility concerns than older physicians that is also in line with the recommendation of the International PCOS guideline [16].

Previous studies, ours included, have found that depression and anxiety [6, 17] and psychological stress [18] are more prevalent in women with PCOS even beyond fertile age compare to non-PCOS counterparts. Moreover, women with PCOS have poorer quality of life than women without the syndrome [19]. Even though the data on mental health is not new, the current study shows that physicians are not well aware of coexistence of depression and anxiety and reduced quality of life in women with PCOS. This is in line with previous studies. Indeed,

**Table 4. Treatments most commonly prescribed for non-fertility-related and fertility-related PCOS concerns.**

| Characteristic | Overall | Denmark | Estonia | Finland | Iceland | Norway | Sweden | P |
|---|---|---|---|---|---|---|---|---|
| *Treatments most commonly prescribed for nonfertility concerns (N = 379)* | | | | | | | | |
| Anti-androgens | 12 | 6 | 27 | 10 | 8 | 15 | 13 | 0.02 |
| Laser depilation | 6 | 15 | 0 | 3 | 0 | 7 | 8 | 0.005 |
| Lifestyle modifications | 66 | 75 | 59 | 63 | 62 | 72 | 58 | 0.31 |
| Metformin | 45 | 59 | 65 | 36 | 54 | 48 | 21 | <0.001 |
| Oral contraceptives | 72 | 82 | 76 | 71 | 92 | 59 | 63 | 0.02 |
| *Treatments most commonly prescribed for fertility concerns (N = 361)* | | | | | | | | |
| Clomiphene citrate | 32 | 18 | 29 | 37 | 31 | 31 | 41 | 0.11 |
| Clomiphene citrate with metformin | 29 | 11 | 57 | 28 | 8 | 31 | 23 | <0.001 |
| Lifestyle modifications | 56 | 69 | 49 | 52 | 46 | 61 | 50 | 0.16 |
| Metformin | 36 | 53 | 37 | 31 | 38 | 39 | 14 | 0.01 |
| Ovulation inductors | 25 | 15 | 29 | 24 | 77 | 12 | 55 | <0.001 |

**Table 5. Multivariable models on the associations of physician characteristics with PCOS knowledge and practices.**

| Characteristic | Awareness of symptom improvement with weight loss | Estimated national PCOS prevalence > 10% | Using Rotterdam diagnostic criteria | Recommend OCP | Recommend lifestyle management for nonfertility concerns | Recommend metformin for nonfertility concerns | Recommend lifestyle management for fertility concerns | Recommend clomiphene citrate | Recommend metformin plus clomiphene citrate |
|---|---|---|---|---|---|---|---|---|---|
| **Sex** | | | | | | | | | |
| Men | 1 | 1 | 1 | 1 | 1 | 1 | 1 | 1 | 1 |
| Women | 1.10 (0.50–2.42) | 2.62 (1.45–4.74) | 0.73 (0.40–1.34) | 0.93 (0.51–1.67) | 1.69 (0.96–2.99) | 1.21 (0.69–2.13) | 0.98 (0.56–1.69) | 0.71 (0.38–1.32) | 0.97 (0.52–1.84) |
| **Age** | | | | | | | | | |
| ≥35 | 1.62 (0.66–4.00) | 2.06 (1.09–3.91) | 1.53 (0.76–3.05) | 0.98 (0.46–2.06) | 0.91 (0.46–1.81) | 1.69 (0.88–3.23) | 2.20 (1.13–4.30) | 0.64 (0.30–1.33) | 1.41 (0.70–2.86) |
| 36–45 | 1.12 (0.52–2.40) | 1.27 (0.71–2.27) | 1.55 (0.84–2.86) | 1.02 (0.53–1.99) | 0.58 (0.32–1.06) | 1.18 (0.67–2.06) | 1.10 (0.62–1.94) | 0.69 (0.37–1.28) | 0.79 (0.41–1.53) |
| 46–55 | 1.34 (0.60–3.01) | 1.08 (0.58–1.99) | 1.11 (0.59–2.10) | 0.41 (0.22–0.76) | 1.12 (0.59–2.11) | 1.00 (0.54–1.85) | 1.07 (0.59–1.93) | 1.03 (0.55–1.93) | 0.93 (0.47–1.83) |
| ≥56 | 1 | 1 | 1 | 1 | 1 | 1 | 1 | 1 | 1 |
| **Specialty** | | | | | | | | | |
| OBGYN/ RE | 1 | 1 | 1 | 1 | 1 | 1 | 1 | 1 | 1 |
| ME | 3.23 (1.09–9.59) | 1.55 (0.80–3.01) | 1.03 (0.53–2.00) | 0.47 (0.24–0.89) | 3.62 (1.62–8.08) | 5.05 (2.51–10.16) | 1.53 (0.78–3.02) | 0.12 (0.04–0.38) | 0.26 (0.09–0.71) |
| **Annual patients with PCOS** | | | | | | | | | |
| <50 | 1 | 1 | 1 | 1 | 1 | 1 | 1 | 1 | 1 |
| ≥50 | 1.93 (0.97–3.85) | 2.48 (1.52–4.06) | 3.05 (1.67–5.58) | 0.80 (0.46–1.37) | 1.41 (0.84–2.38) | 1.19 (0.74–1.93) | 0.90 (0.56–1.45) | 0.76 (0.46–1.28) | 1.14 (0.68–1.91) |

ME, medical endocrinologist; OBGYN, obstetrician-gynaecologist; PCOS, polycystic ovary syndrome; RE, reproductive endocrinologist

Odds ratios adjusted for sex, age, specialty and annual patients with PCOS, and controlled for country as a cluster

Australian primary care physicians did not consider psychological and metabolic comorbidities as clinical features of PCOS [9]. Moreover, North-American obstetrician-gynecologists were less aware of anxiety, depression and reduced quality of life in women with PCOS [8]. In a study conducted among the members of the European Society of Endocrinology [15], 64% of endocrinologists regarded obesity and type 2 diabetes as the primary long-term concerns for PCOS, 20% infertility, 12% cardiovascular diseases, 3% psychological problems and 1% considered endometrial cancer. Given all this and the fact that the risk for psychological distress among women with PCOS is 2-fold [6, 20], screening women with PCOS for psychological comorbidities is recommended.

The present study showed inconsistent management of PCOS across the Nordic countries and Estonia. Gaps in physicians' management of PCOS have also been reported in other studies [8]. In North America, reproductive endocrinologists recommend lifestyle changes for management of PCOS more often than obstetrician-gynecologists, whereas we found that younger doctors were more likely to offer lifestyle management compared to older colleagues. Also discipline differences were noted as medical endocrinologists were more likely to prescribe metformin than OCPs, although they are not mutually exclusive as suggested by the new PCOS guideline. The choice of treatment for the health care professional is likely influenced by the symptom the woman deems most crucial and concerning but also by the awareness and updates available of the current treatment guidelines. The current questionnaire was

fulfilled just before the PCOS guideline was launched, and the results indicate that the health professionals would benefit from getting more information and education. The new international PCOS guideline and the implementations process aims to improve these aspects [14].

## Limitations

The current research was a multinational and multi-disciplinary study and used a novel questionnaire for a common syndrome. However, the number of physicians who took part in this study particularly in Sweden, was small, and the study had low statistical power to estimate reliably the physicians' knowledge and management of PCOS in each country. The number of respondents from Sweden was smaller than expected, whereas the number of targeted physicians in Iceland is small altogether. Finnish physicians participated in this study more than physicians of other Nordic countries. The study was conducted by a Finnish research group, which explains the larger number of Finnish participants. Due to a small number of reproductive endocrinologists, we combined obstetricians/gynecologists and reproductive endocrinologists into a single group in the multivariable models. In the Nordic countries, a reproductive endocrinologist is a gynaecologist with additional training in infertility treatment. In Denmark, Finland and Sweden, reproductive endocrinology is recognized as a subspecialty of gynecology. In the current study, reproductive endocrinologists more often used Rotterdam diagnostic criteria than obstetricians/gynecologists (adjusted OR = 7.3, CI 1.7–31.5). However, other PCOS knowledge and practices did not differ between obstetricians/gynecologists and reproductive endocrinologists. Taking this into account, the findings may not represent obstetrician-gynecologists' and endocrinologists' awareness, diagnosis and management of PCOS in the entire country or within the disciplinary, but offers an overview of PCOS awareness in the Nordic countries and Estonia. It is possible that the results are also affected by a selection bias as the health professionals that are aware of the syndrome are more likely to answer the questionnaire. If this would be the case it would underline the need for more information and the new international PCOS guideline. The questionnaire also lacked the questions on the use of insulin sensitizers to reduce insulin resistance and aromatase inhibitors to induce ovulation. Insulin resistance and compensatory hyperinsulinemia are present in women with PCOS and insulin sensitizing drugs such as inositols are effective in improving PCOS symptoms [21, 22].

## Conclusions

The findings of the present study suggest that the obstetrician-gynecologists, reproductive and medical endocrinologists in Nordic countries and Estonia do not consistently use Rotterdam diagnostic criteria and are not fully aware of some common comorbidities associated with PCOS, particularly psychosocial comorbidities. Furthermore, the management of PCOS seemed to be inconsistent between different physician groups. Considering these and other findings internationally, doctors need more information and education on PCOS. For universal diagnosis and treatment of PCOS, the recently published international PCOS guidelines are well needed and welcomed. Future efforts should be made to increase the awareness of the guidelines and to promote implementation into practice.

## Author Contributions

**Conceptualization:** Terhi T. Piltonen.

**Data curation:** Terhi T. Piltonen, Helle Karro, Linda Kujanpää, Laure Morin-Papunen, Elisabet Stener-Victorin, Inger Sundrström-Poromaa, Angelica L. Hirschberg, Pernille Ravn,

Dorte Glintborg, Jan Roar Mellembakken, Thora Steingrimsdottir, Melanie Gibson-Helm, Eszter Vanky, Marianne Andersen, Riikka K. Arffman, Helena Teede.

**Formal analysis:** Kobra Falah-Hassani.

**Funding acquisition:** Terhi T. Piltonen.

**Project administration:** Terhi T. Piltonen.

**Supervision:** Terhi T. Piltonen.

**Writing – original draft:** Terhi T. Piltonen, Maria Ruokojärvi, Kobra Falah-Hassani.

**Writing – review & editing:** Terhi T. Piltonen, Maria Ruokojärvi, Helle Karro, Linda Kujan-pää, Laure Morin-Papunen, Juha S. Tapanainen, Elisabet Stener-Victorin, Inger Sundr-ström-Poromaa, Angelica L. Hirschberg, Pernille Ravn, Dorte Glintborg, Jan Roar Mellembakken, Thora Steingrimsdottir, Melanie Gibson-Helm, Eszter Vanky, Marianne Andersen, Riikka K. Arffman, Helena Teede, Kobra Falah-Hassani.

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
