## [Decision Letter · Decision Letter 0]

20 Sep 2019

PONE-D-19-23460

Awareness of polycystic ovary syndrome among obstetrician-gynecologists and endocrinologists in Northern Europe

PLOS ONE

Dear Dr Falah-Hassani,

Thank you for submitting your manuscript to PLOS ONE. After careful consideration, we feel that it has merit but does not fully meet PLOS ONE’s publication criteria as it currently stands. Therefore, we invite you to submit a revised version of the manuscript that addresses the points raised during the review process.

We would appreciate receiving your revised manuscript by Nov 04 2019 11:59PM. To enhance the reproducibility of your results, we recommend that if applicable you deposit your laboratory protocols in protocols.io, where a protocol can be assigned its own identifier (DOI) such that it can be cited independently in the future. For instructions see: http://journals.plos.org/plosone/s/submission-guidelines#loc-laboratory-protocols

We look forward to receiving your revised manuscript.

Kind regards,

Antonio Simone Laganà, M.D.

Academic Editor

PLOS ONE

Additional Editor Comments:

The reviewers have expressed positive comments regarding your article, raising only few concerns. Considering this point, I invite authors to perform the required minor revisions.

2. Please include additional information regarding the survey or questionnaire used in the study and ensure that you have provided sufficient details that others could replicate the analyses. For instance, if you developed a questionnaire as part of this study and it is not under a copyright more restrictive than CC-BY, please include a copy, in both the original language and English, as Supporting Information.  If the original language is written in non-Latin characters, for example Amharic, Chinese, or Korean, please use a file format that ensures these characters are visible

3. Please state whether you validated the questionnaire prior to testing on study participants. Please provide details regarding the validation group within the methods section.

"The authors declare that they have no financial disclosures. "

Please provide an amended Funding Statement that declares *all* the funding or sources of support received during this specific study (whether external or internal to your organization) as detailed online in our guide for authors at http://journals.plos.org/plosone/s/submit-nowPlease state what role the funders took in the study.  If any authors received a salary from any of your funders, please state which authors and which funder. If the funders had no role, please state: "The funders had no role in study design, data collection and analysis, decision to publish, or preparation of the manuscript."

Reviewers' comments:

Reviewer's Responses to Questions

**Comments to the Author**

1. Is the manuscript technically sound, and do the data support the conclusions?

Reviewer #1: Yes

Reviewer #2: Yes

Reviewer #3: Yes

2. Has the statistical analysis been performed appropriately and rigorously? 

Reviewer #1: Yes

Reviewer #2: Yes

Reviewer #3: Yes

3. Have the authors made all data underlying the findings in their manuscript fully available?

Reviewer #1: No

Reviewer #2: Yes

Reviewer #3: Yes

4. Is the manuscript presented in an intelligible fashion and written in standard English?

Reviewer #1: Yes

Reviewer #2: Yes

Reviewer #3: Yes

5. Review Comments to the Author

Reviewer #1: Falah-Hassani and colleagues report on the clinical knowledge of 382 physicians from Finland, Denmark, Norway,

Estonia, and Sweden or Iceland.

1) The investigators note that Finland, Denmark, Norway, Sweden and Iceland) and Estonia share similar health care systems (Holm et al. Health Care Anal 1999;7:321-30). Are there any other reasons to aggregate these countries?

2) Overall, 43% of respondents were from Finland, much higher than the rest of the countries. How did this bias the results?

3) 75% of respondents were obstetrician-gynecologists. What was the power to detect trends for reproductive endocrinologists (which the investigators mention in their discussion of limitations) medical endocrinologists? Pediatric endocrinologists? General practitioners?

4) The questionnaire was part of larger international study conducted to inform translation needs for the new international PCOS guidelines published in 2018 (Gibson-Helm et al. Semin Reprod Med 2018;36:19-27). How do these data differ from the aggregated original data?

Minor:

a) Please include the questionnaire used.

b) Fig. 1 is difficult to interpret readily. Suggest a graph with 4 separate bar graphs, or better stil 'Box and Whisker Plot'.

c) It is unclear whether the data of this study being made available?

Reviewer #2: I was pleased to revise the manuscript entitled “Awareness of polycystic ovary syndrome among obstetrician-gynecologists and endocrinologists in Northern Europe” (Manuscript Number: PONE-D-19-23460).

The study was approved by the Institutional Animal Care and Use Committee from Ponce Health Sciences University protocol #202 and from the University of Texas at rio Grande Valley protocol #2016-004. In general, this manuscript was aimed to investigate the physician reported awareness, diagnosis and management of PCOS and to explore the differences between medical disciplines in the Nordic countries and Estonia. In my honest opinion, the topic is interesting enough to attract the readers’ attention. Methodology is accurate and conclusions are supported by the data analysis. Nevertheless, authors should clarify some point and improve the discussion citing relevant and novel key articles about the topic.

In general, the Manuscript may benefit from several minor revisions, as suggested below:

1. Abstract. I would suggest improving description of study design, the use of a survey is missed.

2. Methods. I would suggest providing further information regarding the questionnaire development.

3. Methods. How the surgery results were evaluated and introduced in the analysis? In example how the knowledge of POCS comorbidities was evaluated?

4. Accumulating evidence suggests that one of the most important mechanisms of PCOS pathogenesis is the insulin-resistance. For this reason, the use of insulin-sensitizers, such an inositol isoform, gained increasing attention due to their safety profile and effectiveness. Authors may better discuss this point, taking to account these recent articles: PMID: 30270194; PMID: 28835764; PMID: 30538744; PMID: 27737594.

Reviewer #3: The authors investigated in the present manuscript the "Awareness of polycystic ovary syndrome among obstetrician-gynecologists and endocrinologists in Northern Europe". The topic is of scientific importance and deserves publication in your journal. It is generally well written with clear methodology and transparent results. The discussion is appropriately written and the limitations of the study are accurately presented. I believe that the manuscript would only benefit from a supplemental file (or a link) that would provide the actual questionnaire which could be used by future studies in this field.

6. PLOS authors have the option to publish the peer review history of their article (what does this mean?). If published, this will include your full peer review and any attached files.

Reviewer #1: No

Reviewer #2: No

Reviewer #3: No

---

## [Author Response · Author response to Decision Letter 0]

15 Oct 2019

Response to Reviewers

Thank you for the thoughtful comments to improve the manuscript. We took into consideration all comments made by the editor and reviewers, and revised the paper accordingly. Below we explain how we have addressed with each of the comments. Modifications in the manuscript are highlighted. 

Editor comments:

The reviewers have expressed positive comments regarding your article, raising only few concerns. Considering this point, I invite authors to perform the required minor revisions.

Response: The manuscript meets PLOS ONE's style requirements.

2. Please include additional information regarding the survey or questionnaire used in the study and ensure that you have provided sufficient details that others could replicate the analyses. For instance, if you developed a questionnaire as part of this study and it is not under a copyright more restrictive than CC-BY, please include a copy, in both the original language and English, as Supporting Information. If the original language is written in non-Latin characters, for example Amharic, Chinese, or Korean, please use a file format that ensures these characters are visible

Response: We have provided a link to the study original questionnaire.

https://www.fertstert.org/article/S0015-0282(17)30344-8/addons

The slightly modified version used in the present study is attached. 

We have also added further information about the survey questionnaire on page 6, paragraph 1.

3. Please state whether you validated the questionnaire prior to testing on study participants. Please provide details regarding the validation group within the methods section.

Response: Validation may not be relevant for the whole questionnaire. For instance, no validation is required or is appropriate for assessing physicians’ knowledge and their support needs. The questions collect data on participants’ perceptions only. We agree that some parts regarding physicians’ perceptions of the care they provide and comorbidities may need validation. They are not the same as conducting an audit of medical records. However, this type of questionnaire is a more appropriate method for the aims of this study. Asking for healthcare providers perceptions of the care they provide tells us more about their knowledge of what care they should be providing. This links well with the other sections about knowledge of the condition and support needs and is more suitable for a study aiming to inform knowledge translation activities for healthcare providers. The questionnaire used here is adapted from questionnaires previously published in high-quality peer-reviewed literature, which also required no validation studies, enabling comparison to, and build on, prior knowledge in this area.

"The authors declare that they have no financial disclosures. "

a. Please provide an amended Funding Statement that declares *all* the funding or sources of support received during this specific study (whether external or internal to your organization) as detailed online in our guide for authors at http://journals.plos.org/plosone/s/submit-now

b. Please state what role the funders took in the study. If any authors received a salary from any of your funders, please state which authors and which funder. If the funders had no role, please state: "The funders had no role in study design, data collection and analysis, decision to publish, or preparation of the manuscript."

Response: This study received no funding. We have added “Funding Statement” to the manuscript on page 19.

Reviewers' comments:

Reviewer's Responses to Questions

Comments to the Author

1. Is the manuscript technically sound, and do the data support the conclusions?

Reviewer #1: Yes

Reviewer #2: Yes

Reviewer #3: Yes

2. Has the statistical analysis been performed appropriately and rigorously? 

Reviewer #1: Yes

Reviewer #2: Yes

Reviewer #3: Yes 

3. Have the authors made all data underlying the findings in their manuscript fully available?

Reviewer #1: No

Reviewer #2: Yes

Reviewer #3: Yes 

Response: The data is available upon request.

4. Is the manuscript presented in an intelligible fashion and written in standard English?

Reviewer #1: Yes

Reviewer #2: Yes

Reviewer #3: Yes ________________________________________

5. Review Comments to the Author

Reviewer #1: Falah-Hassani and colleagues report on the clinical knowledge of 382 physicians from Finland, Denmark, Norway, Estonia, and Sweden or Iceland.

1) The investigators note that Finland, Denmark, Norway, Sweden and Iceland) and Estonia share similar health care systems (Holm et al. Health Care Anal 1999;7:321-30). Are there any other reasons to aggregate these countries?

Response: We have reported other similarities between the Nordic countries and Estonia on page 5, paragraph 1.

2) Overall, 43% of respondents were from Finland, much higher than the rest of the countries. How did this bias the results?

Response: The reason for high participation rate in Finland was the fact that the study was initiated by Finnish investigators. The sample size is too small to run any multivariable model for each country. We think that the data is valuable showing that even in countries with well-organized health care there is a need to increase awareness of PCOS and the related comorbidities. More suitable materials and education were also lacking in these countries. 

3) 75% of respondents were obstetrician-gynecologists. What was the power to detect trends for reproductive endocrinologists (which the investigators mention in their discussion of limitations) medical endocrinologists? Pediatric endocrinologists? General practitioners?

Response: Thank you for this important question. We have not included pediatric endocrinologists and general practitioners in the current study. This study did not have statistical power to detect the differences in PCOS awareness among medical endocrinologists or reproductive endocrinologists for each individual country.

4) The questionnaire was part of larger international study conducted to inform translation needs for the new international PCOS guidelines published in 2018 (Gibson-Helm et al. Semin Reprod Med 2018;36:19-27). How do these data differ from the aggregated original data?

Response: We have clarified the differences on page 6, paragraph 1.

Minor:

a) Please include the questionnaire used.

Response: We have provided a link to the original study questionnaire and attached the slightly modified version of the questionnaire here. 

https://www.fertstert.org/article/S0015-0282(17)30344-8/addons

b) Fig. 1 is difficult to interpret readily. Suggest a graph with 4 separate bar graphs, or better stil 'Box and Whisker Plot'.

Response: Thank you for this comment. We have now changed Figure 1 to a graph with four separate bar graphs.

c) It is unclear whether the data of this study being made available?

Response: The data of this survey is available upon request.

Reviewer #2: 

I was pleased to revise the manuscript entitled “Awareness of polycystic ovary syndrome among obstetrician-gynecologists and endocrinologists in Northern Europe” (Manuscript Number: PONE-D-19-23460).

The study was approved by the Institutional Animal Care and Use Committee from Ponce Health Sciences University protocol #202 and from the University of Texas at rio Grande Valley protocol #2016-004. In general, this manuscript was aimed to investigate the physician reported awareness, diagnosis and management of PCOS and to explore the differences between medical disciplines in the Nordic countries and Estonia. In my honest opinion, the topic is interesting enough to attract the readers’ attention. Methodology is accurate and conclusions are supported by the data analysis. Nevertheless, authors should clarify some point and improve the discussion citing relevant and novel key articles about the topic.

Response: Thank you for your supportive comments. We have addressed all your comments and revised the manuscript accordingly. 

In general, the Manuscript may benefit from several minor revisions, as suggested below:

1. Abstract. I would suggest improving description of study design, the use of a survey is missed.

Response: We have added “survey” to the methods section of the abstract.

2. Methods. I would suggest providing further information regarding the questionnaire development.

Response: We have added further information about the survey questionnaire on page 6, paragraph 1.

3. Methods. How the surgery results were evaluated and introduced in the analysis? In example how the knowledge of POCS comorbidities was evaluated?

Response: We have reported the results for surgery for ovarian cysts on page 11, paragraph 1. All the results were based on the physicians’ own experiences and knowledge. We have not validated the data on surgery. 

4. Accumulating evidence suggests that one of the most important mechanisms of PCOS pathogenesis is the insulin-resistance. For this reason, the use of insulin-sensitizers, such an inositol isoform, gained increasing attention due to their safety profile and effectiveness. Authors may better discuss this point, taking to account these recent articles: PMID: 30270194; PMID: 28835764; PMID: 30538744; PMID: 27737594.

Response: We have discussed the use of insulin-sensitizers on page 17.

Reviewer #3: 

The authors investigated in the present manuscript the "Awareness of polycystic ovary syndrome among obstetrician-gynecologists and endocrinologists in Northern Europe". The topic is of scientific importance and deserves publication in your journal. It is generally well written with clear methodology and transparent results. The discussion is appropriately written and the limitations of the study are accurately presented. I believe that the manuscript would only benefit from a supplemental file (or a link) that would provide the actual questionnaire which could be used by future studies in this field.

Response: Thank you for your support and comments. We have provided a link to the survey questionnaire. 

https://www.fertstert.org/article/S0015-0282(17)30344-8/addons

---

## [Decision Letter · Decision Letter 1]

20 Nov 2019

Awareness of polycystic ovary syndrome among obstetrician-gynecologists and endocrinologists in Northern Europe

PONE-D-19-23460R1

Dear Dr. Piltonen,

We are pleased to inform you that your manuscript has been judged scientifically suitable for publication and will be formally accepted for publication once it complies with all outstanding technical requirements.

With kind regards,

Antonio Simone Laganà, M.D.

Academic Editor

PLOS ONE

Additional Editor Comments (optional):

Authors performed the required corrections, which were positively evaluated by the reviewers. I am pleased to accept this paper for publication.

Reviewers' comments:

Reviewer's Responses to Questions

**Comments to the Author**

1. If the authors have adequately addressed your comments raised in a previous round of review and you feel that this manuscript is now acceptable for publication, you may indicate that here to bypass the “Comments to the Author” section, enter your conflict of interest statement in the “Confidential to Editor” section, and submit your "Accept" recommendation.

Reviewer #1: All comments have been addressed

Reviewer #2: All comments have been addressed

2. Is the manuscript technically sound, and do the data support the conclusions?

Reviewer #1: Yes

Reviewer #2: Yes

3. Has the statistical analysis been performed appropriately and rigorously? 

Reviewer #1: Yes

Reviewer #2: Yes

4. Have the authors made all data underlying the findings in their manuscript fully available?

Reviewer #1: Yes

Reviewer #2: Yes

5. Is the manuscript presented in an intelligible fashion and written in standard English?

Reviewer #1: Yes

Reviewer #2: Yes

6. Review Comments to the Author

Reviewer #1: The authors have responded to the extent possible to all prior queries. How to obtain the data should be made more clearly.

Reviewer #2: I was pleased to revise the manuscript entitled “Awareness of polycystic ovary syndrome among obstetrician-gynecologists and endocrinologists in Northern Europe” (Manuscript Number: PONE-D-19-23460).

This manuscript was aimed to investigate the physician reported awareness, diagnosis and management of PCOS and to explore the differences between medical disciplines in the Nordic countries and Estonia. In my honest opinion, the topic is interesting enough to attract the readers’ attention. Methodology is accurate and conclusions are supported by the data analysis. Moreover, the authors performed all the suggested revisions and I appreciated the manuscript improvement.

7. PLOS authors have the option to publish the peer review history of their article (what does this mean?). If published, this will include your full peer review and any attached files.

Reviewer #1: No

Reviewer #2: No

---

## [Editor Report · Acceptance letter]

16 Dec 2019

PONE-D-19-23460R1 

Awareness of polycystic ovary syndrome among obstetrician-gynecologists and endocrinologists in Northern Europe 

Dear Dr. Piltonen:

I am pleased to inform you that your manuscript has been deemed suitable for publication in PLOS ONE. Congratulations! Your manuscript is now with our production department. 

With kind regards,

on behalf of

Dr. Antonio Simone Laganà 

Academic Editor

PLOS ONE